# Germinated Wheat as a Potential Natural Source of Antioxidants to Improve Sperm Quality: A Canary Trial

**DOI:** 10.3390/vetsci11010004

**Published:** 2023-12-20

**Authors:** Arda Onur Özkök, Gözde Kilinç

**Affiliations:** 1Department of Veterinary, Suluova Vocational School, Amasya University, 05500 Amasya, Türkiye; 2Department of Food Processing, Suluova Vocational School, Amasya University, 05500 Amasya, Türkiye; gozde.kilinc@amasya.edu.tr

**Keywords:** antioxidant, canary, germinated wheat, spermatological parameters

## Abstract

**Simple Summary:**

Canaries, which have various songs and colors, play an important role in the pet industry as cage birds. Artificial insemination is a very important process to provide canaries with the desired genetic characteristics for the sector. Therefore, sperm quality should be improved. Improving sperm quality mainly depends on the preservation of lipids in the spermatozoa structure from oxidation. The lipids in the spermatozoa cell membrane can be easily oxidized, which can negatively affect the quality of the sperm. Natural and synthetic antioxidants are used to prevent lipid oxidation. In recent years, natural antioxidants have attracted considerable attention. In this study, the in vitro antioxidant content of wheat, which was germinated at different times (first, third, and fifth days), was determined, and sperm quality was examined by feeding this germinated wheat (germinated for 5 days) to Gloster canaries. It was revealed that the motility (%) and viability rates (%) increased in sperm samples obtained from canaries fed using germinated wheat. Given the results achieved, it is thought that germinated wheat can be used as a natural antioxidant source to improve some sperm quality parameters in canaries.

**Abstract:**

The present study was carried out to determine the effects of germinated wheat on some spermatological parameters (motility, abnormal spermatozoa, hypo-osmotic swelling test (HOST), and viability) in male Gloster canaries. For this purpose, the canaries were divided into two groups, each consisting of six canaries, one control (C), and the other experimental group (GW). Group C was fed commercial canary feed. The GW group was fed germinated wheat (germinated for 5 days) and commercial canary feed. The canaries were kept individually in four-storey cages (60 cm × 50 cm × 40 cm). In this study, which lasted 2 months, feed/water was provided ad libitum and lighting was applied daily for 16 h (turned on at 05:30 and turned off at 21:30). At the end of the experiment, the sperm samples obtained from the canaries were examined in terms of the aforementioned parameters. The effects of germination time × concentration, germination time, and solvent rate on DPPH radical scavenging activity and phenolic compounds in germinated wheat were significant (*p* < 0.001). The highest DPPH (2, 2-diphenyl-1-picryl hydrazyl) radical scavenging activity (86.06%) and phenolic content (316.25 mg GAE/g) in germinated wheats were obtained on the fifth day (90% solvent rate) of germination. It was determined that germinated wheat did not significantly affect abnormal sperm and HOST values (*p* > 0.05). However, it significantly affected the motility and viable sperm rates (*p* < 0.05). As a result, it is thought that germinated wheat can be used as a natural antioxidant source to increase motility and vitality in canary sperm.

## 1. Introduction

Oxidative stress, which leads to cell death or aging due to cellular damage, is caused by reactive oxygen species [1]. Even though reactive oxygen species (ROS) play an important role during spermatozoon production in animal organisms, high ROS levels damage spermatozoa due to lipid peroxidation [2]. In addition to inhibiting the fertility of spermatozoa, they can also negatively affect the health of offspring by causing the degradation of lipids, proteins, and nucleic acids [3]. Aerobic organisms produce various enzymes to prevent these harmful effects of ROS [4,5]. However, these enzymes might be insufficient in some cases [6]. Therefore, antioxidant supplementation is carried out to diet in poultry [7]. Antioxidants decelerate lipid oxidation and improve sperm quality [8]. Various natural and synthetic antioxidants were used for this purpose [9,10].

The demand for natural antioxidant sources has increased in recent years [11,12]. One of the natural antioxidant sources is plants [13,14]. Plants play an important role with the secondary metabolites that contain [15,16]. Previous studies [17,18,19,20,21,22] investigated various antioxidants to improve sperm quality in different animal species.

It is known that grains such as wheat, corn, barley, and rye have high vitamin E content, which is a natural antioxidant [23]. It was reported that some chemical properties of cereal grains can be improved, and their functionality increases with germination [24]. In many studies [25,26,27], it was stated that the antioxidant capacity of germinated wheat increased.

In the literature review, the number of studies on canary sperm was determined to be limited, and no studies on germinated wheat could be found. It was reported that the wheat sprout improves sperm quality parameters in rats [28,29] and rabbits [30].

The present study investigated the antioxidant traits of germinated wheat and the effects of germinated wheat seeds on some spermatological parameters (motility, abnormal sperm, hypo-osmotic swelling Test, and viability) in Gloster canaries.

## 2. Materials and Methods

### 2.1. Ethics Statement

The Ethics Committee approval for this study was obtained from Ondokuz Mayıs University Animal Experiments Local Ethics Committee (No: 2022/38).

### 2.2. Preparation of Wheat Germination

Wheat (*Triticum aestivum* L.) materials were germinated following the method introduced by Kılınçer and Demir [24]. Accordingly, wheat (Kunduru-1149 durum wheat genotype) seeds were washed with distilled water and filtered using a strainer. To prevent possible microbial activity during germination, the seeds were kept in a 15% sodium hypochlorite solution for 25 min and then passed through distilled water again. Distilled water was added to the sterilized seeds. Approximately 3 times the amount of the material was added. The materials were kept in the dark at room temperature for 12 h. After this process, they were washed again, passed through water, and filtered. The germination process was then completed. In this study, wheat seeds were germinated for 1, 3, and 5 days. The GW group was fed with a 5-day germinated wheat supplement. The in vitro antioxidant content of wheat, which was germinated at different times (first, third, and fifth days), was determined.

### 2.3. Determination of Antioxidant Activity of Germinated Wheat

Wheat samples germinated at different times were dried in an oven (40 °C) for 24–48 h and ground into powder using a blender. Germinated wheat powder was added to 1/30 ethanol solution (30, 60, and 90%). The extraction process was performed in an ultrasonic bath at 30–40 °C for 1 h. The mixtures removed from the ultrasonic bath were filtered using ordinary filter paper. The solvent was removed from the mixtures in a water bath (50 °C) to prepare liquid extracts. Antioxidant activity in germinated wheat was examined with 4 replications.

DPPH (2, 2-diphenyl-1-picryl-hydrazyl) radical scavenging activity and total phenolic content of wheat germinated for different periods (1, 3, and 5 days) were then determined.

The method introduced by Blois [31] was used by modifying for DPPH reducing power (%) analysis. Accordingly, 0.2 mL of the prepared extracts was transferred to glass tubes, and 3 mL of ethanol was added. Then, 375 µL of 1 mM DPPH solution (prepared with ethanol) was added to the mixtures. The tubes were mixed for 5–10 s using a vortex and then incubated in the dark for 30 min. The absorbance values of the control and samples were scanned using a spectrophotometer at a wavelength of 517 nm, and the DPPH radical scavenging activity (%) of the extracts was calculated using the following formula [32].
DPPH (%) = (Control absorbance-Sample Absorbance)/Control absorbance × 100
(Control: free-extract; Blank: ethanol)

The method introduced by Eser et al. [33] was used for total phenolic content analysis. Accordingly, 50 µL samples were obtained from each extract and dissolved by adding 950 µL of distilled water; then, 2.5 mL of 2% sodium carbonate was added. Subsequently, 500 µL Folin–Ciocalteu reagent was added, and the tubes were vortexed. The samples were kept in the dark for 40 min. Then, the absorbances were measured by using a spectrophotometer at a wavelength of 725 nm.

### 2.4. Experimental Plan and Feeding of Canaries

The canary (*Serinus canaria*) experiment was carried out in 4-layer cages (60 cm × 50 cm × 40 cm) at the Suluova Vocational School Laboratory of Amasya University. The canaries were divided into 2 groups, with 6 male Gloster canaries at the age of 2–3 years in each group (Table 1).

Group C was fed only the commercial canary feed (Crude protein content: 18.5%), whereas the GW group was fed the commercial canary feed and a 5-day germinated wheat supplement. The germinated wheat was given freshly to the experimental group. The chemical composition of germinated wheat, which was analyzed following AOAC [34], is presented in Table 2.

The canary experiment lasted for 2 months. During the process, feed and water were provided ad libitum. The photoperiod applied was 16 h of light and 8 h of darkness per day (turned on at 05:30 and turned off at 21:30).

### 2.5. Collecting Sperm from Canaries

The homogeneity of control and experiment group, in terms of spermatological parameters, was tested prior to the experiment (*p* > 0.05).

Before starting the study, the canaries were accustomed to giving sperm. During the acclimatization process, sperm was collected from the birds once every 4 days. Before collecting sperm samples from canaries, the lighting period was gradually increased to 16 h [35]. Thus, they were able to become sexually active. In birds, the testicles are not outside the body as in mammals but are located in the coelomic space in the abdomen [36]. Therefore, sperm samples were collected from the canaries using the cloacal massage method [37]. An average amount of 2 µL of sperm was taken from each canary. Contaminated sperm samples were excluded.

### 2.6. The Evaluation of Spermatological Parameters

A total of 4 spermatologic parameters, including motility (%), HOST (hypo-osmotic swelling test; %), abnormal sperm ratio (%), and viability ratio (%), were evaluated in this study. 

The sperm samples obtained from the canaries at the end of the experiment were examined without any delay. DMEM diluter was used while analyzing the fresh sperm motility parameter. 

The method introduced by Laskemoen et al. [38] was used to determine the motility rate. Sperm samples diluted with DMEM were examined using a heated (40 °C), plate-supported BRESSER Researcher LCD microscope. Forward motile spermatozoa movement was recorded for 1.5 min. Motility values of spermatozoa examined at 40× magnification were expressed as % (Figure 1). 

HOST was used to determine the plasma membrane integrity. The method introduced by Ramu and Jeyendran [39] was used to determine the HOST. The fresh sperm samples added into 100 mOsmol HOST solution were kept in a heated water bath at 37 °C for at least 30 min. Then, spermatozoa with swelling and tail curling were evaluated as HOST-positive and presented as % (Figure 2).

The method introduced by Birkhead et al. [40] was used to determine the abnormal sperm count. After the sperm samples were fixed with 5% formalin solution, it was taken on a slide and frotis were drawn. The sperm was fixed and then stained with Giemsa for 15 min. Then, it was examined using a light microscope at 60× magnification. A total of 200 spermatozoa were counted. Anomalies were detected in the head, acrosome, middle part, and tail (Figure 3).

The method introduced by Fischer et al. [41] was used for the viability rate. The diluted fresh sperm was mixed 1/2 with eosin solution prepared with 2% and 3% sodium citrate. Then, a smear was drawn on the slide, dried quickly with air, and examined under a light microscope at 40× magnification. Non-stained sperms were considered viable (Figure 4).

### 2.7. Statistical Analysis

The number of animals used in the present study was determined by using the G*Power (Version, 3.1.4) package program. Independent samples *t*-tests were used to statistically evaluate the spermatological parameters in this study. Two-way ANOVA was performed for antioxidant activity of germinated wheat. The differences between groups were determined by Duncan’s multiple range tests. SPSS 22.0 package program was used for this purpose [42].

## 3. Results

### 3.1. Antioxidant Activity of Germinated Wheats

The DPPH radical scavenging activity (%) and phenolic substance content (mg GAE/g) of germinated wheat are presented in Table 3.

The effect of germination time × concentration, germination time, and solvent rate on DPPH radical scavenging activity % and phenolic compounds was significant (*p* < 0.001). The highest DPPH reduction % value (86.06%) and total phenolic content (316.25 mg GAE/g) were observed on the fifth day of germination and in the extracts prepared with 90% ethanol. 

According to the germination time, the highest DPPH radical scavenging activity was determined on the third day (71.63%), and total phenolic content (281.417 mg GAE/g) was determined on the fifth day (281.417 mg GAE/g). When the groups were compared in terms of concentration, the highest DPPH radical scavenging activity (78.79%) was observed in the extraction with 90% ethanol. According to the solvent rate, the highest total phenolic content (243.417 mg GAE/g) was observed in the 60% ethanol extraction.

### 3.2. Effects on the Spermatological Parameters of Germinated Wheat

The effects of germinated wheat on some spermatological parameters in canaries are shown in Table 4.

Table 4 shows that the effect of germinated wheat on motility and live spermatozoa rate was found to be statistically significant (*p* < 0.05). When compared to the control group, motility (55.83%) and survival rate (59.17%) were higher in canaries fed using germinated wheat. It is thought that this positive effect is related to the high antioxidant activity of germinated wheat.

## 4. Discussion

In this study, it was determined that DPPH radical scavenging activity and phenolic substance content increased as the germination time of wheat increased. In many studies [25,26,27], it was stated that the antioxidant capacity of wheat increased with the germination process. These findings are consistent with the results achieved in the present study. Germination of edible seeds is known to be a process that improves the nutritional value of raw seeds and releases a wide range of bioactive compounds offering health benefits [43]. It was reported that free and bound phenolic compounds, which contribute to increased antioxidant capacity in germinated wheat, increase [44]. It was mentioned in a previous study that the increase in amino acid concentration and antioxidant activity in germinated wheat has positive effects on health [45]. The biological activity of vitamin E might be effective against oxidative stress [46]. Sprouted wheat is known to be rich in vitamin E [47]. Protein digestibility was reported to depend on anti-nutritional factors (e.g., trypsin inhibitors) in foods. It is known that the protein content in germinated seeds is higher when compared to ungerminated seeds. Germination is thought to play a very important role in improving protein digestibility and biological properties of the protein [48]. In addition to mineral components such as zinc and selenium, omega-3 fatty acids and antioxidant vitamins are essential in preventing infertility problems and mainly contribute to minimizing oxidative stress [49]. Germinated seeds were reported to be an excellent source of phenolic antioxidants since phenols synthesized during seed germination can contribute to increased antioxidant activity [50]. It was reported in a previous study that wheat grains steeped for 24 h and germinated for 7 days would produce the most desirable sprouts with respect to antioxidant concentrations and some sensory properties [51].

It is known that wheat (*Triticum aestivum*) is a good source of selenium. In addition, nutritional value increases due to higher vitamin content and better protein quality due to germination and sprouting [52]. Selenium is considered an important component of many enzymes, some of which have antioxidant functions. It sensitizes animals to certain oxidative stress factors due to selenium deficiency [53].

It is known that the vitamin E content of germinated wheat seeds increases during germination [51]. Vitamin E in poultry diet has been shown to significantly protect sperm quality in male birds and egg quality in female birds, reducing lipid peroxidation in sperm and eggs [54].

It was reported that wheat germ contains higher amounts of zinc and manganese in the embryo than other nutrients [55]. It was observed that zinc deficiency has a significant negative effect on sperm quality in poultry breeders. For this reason, it is known that poultry producers pay more attention to organic zinc supplementation in improving reproductive performance, offspring viability, and immune status [56].

The effect of germinated wheat on abnormal sperm and HOST in canaries was determined not to be statistically significant. However, it significantly affected the motility and viability rate (*p* < 0.05). This is thought to be due to the high antioxidant capacity of germinated wheat. 

Oxidative stress prevents the functionality of spermatozoa by negatively affecting sperm membrane permeability [57]. In addition to causing spermatozoa damage, oxidative stress can lead to spermatozoa deformities and infertility problems in males. ROS might also cause excessive energy (ATP) consumption in sperm, resulting in loss of motility and vitality in sperm due to insufficient axonemal phosphorylation and lipid peroxidation [58]. Enzymatic and non-enzymatic antioxidants are thought to increase spermatozoa viability and fertilization ability by acting against ROS and its negative effects on poultry sperm [7]. It was emphasized that additional antioxidants taken with food can improve sperm density, motility, morphological defects, and, sometimes, DNA damage. It was shown that additional antioxidants can significantly improve sperm parameters in infertile men [59]. It was observed that the antioxidant amount of wheat increases strongly during germination. It was observed that antioxidant activity is low in wheat germ and very low in wheat plants [60]. It is known that sprouted wheat has a much higher amount of amino acids than dried wheat. It is also thought that free phenolic compounds increase during germination, causing an increase in the antioxidant capacity of germinated wheat [61]. It was reported that germinated wheat contains higher levels of tocopherols, niacin, and riboflavin. Another benefit of germination is that it increases the levels of free and bound phenolic compounds, which improve antioxidant capacity [62]. It was stated that a significant increase in antioxidant enzyme activities of wheat was observed after 72 h of germination [63]. A previous study reported that antioxidant activity and protein value increased significantly in germinated wheat [26].

Phospholipids in the spermatozoa membrane of birds are rich in PUFA (polyunsaturated fatty acids) [64,65]. PUFA is particularly high in docosatetraenoic (22:4n-6) and arachidonic (20:4n6) [66]. PUFA is necessary to maintain sperm membrane fluidity and the flexibility of sperm motility [67]. However, since these fatty acids are very sensitive to oxidation, they can be easily oxidized and negatively affect normal cell functioning [68]. This phenomenon, which is called oxidative stress, is caused by the disruption of the balance between oxidants and antioxidants [69]. Antioxidant substances play an essential role in preventing oxidative stress [70]. It was reported that these substances are necessary for sperm membrane integrity, motility, and fertilizing ability [71].

A positive relationship was observed between a high antioxidant diet and feeding, reproductive behaviors, and testosterone levels. It was reported that male European starlings (*Sturnus vulgaris*) fed using high-antioxidant diets maintain their frequency of reproductive behaviors. In contrast, those fed using low antioxidant diets demonstrated a decrease in reproductive behaviors depending on environmental conditions [72]. It was observed that male pheasants fed with a diet enriched in antioxidants produce sperm in greater volume and density [73].

As can be seen in the literature review, no studies have investigated the effects of germinated wheat on sperm quality in canaries.

Germinated seeds given to caged birds during incubation can be used as an alter-native to special egg feeds [74].

There are a limited number of studies investigating the effects of wheat sprouts on sperm quality in rats [28,29] and rabbits [30]. A study carried out on rats exposed to lead reported that wheat sprout extract significantly improved sperm quality parameters [29]. In another study carried out on rats exposed to lead, it was determined that wheat sprout extract inhibited oxidative stress in testicular tissues [28]. The results of the present study partially overlap with these findings. A study conducted on rabbits reported that wheat sprouts did not affect some sperm parameters (the morphology of dead spermatozoa, number of motile sperm, acrosome integrity, and sperm dose per ejaculate) [30]. 

In a study examining the effects of supplementary diet and vitamins on sperm parameters in roosters, it was observed that vitamin A supplementation had a positive effect on sperm volume and spermatozoa density, while vitamin E had a greater effect on increasing spermatozoa motility [75]. 

It was reported that vitamin E content also increases with wheat germination [51]. Vitamin E is known to be a natural antioxidant [76].

In the present study, motility and vitality rates were found to be particularly significant in canaries fed using sprouted wheat, which corroborates the previous studies. It is also known that wheat (*Triticum aestivum*) is a good source of selenium, and its vitamin content increases during sprouting [52]. The effectiveness of selenium against oxidative stress was emphasized in animals [53]. For this reason, it is thought to significantly affect spermatozoa viability rate in the present study.

It was observed in a previous study that 20 h of germination of wheat reduced its energy content, but the energy value of wheat was recovered with 48 h of germination [74]. Another striking finding in the study is that broiler chickens’ consumption of wheat germinated for 20 h is significantly less than that of the seeds that were not germinated or germinated for 48 h. In the present study, examining the antioxidant levels of germinated wheat seeds at different times, it was determined that the antioxidant levels were higher on the third day compared to the first day of germination. Since sperm quality was evaluated in the present study, energy rates were not considered. It was determined in study that seeds germinating for 48 h had positive effects on the development of chickens. In the present study, it was observed that germinated wheat seeds had positive effects on sperm motility and viability rates in birds.

In a study examining the effects of oxidative stress on sperm competition in house sparrows, phenotypic characteristics of male birds were observed to exert an effect on spermatozoa morphology and swimming speed, and the relationship with antioxidant release was investigated [77]. The study revealed that male birds that can protect their sperm against oxidative stress elements have better sperm quality. Therefore, it was stated that dominant males, who are more open to a stressful environment, have a decrease in spermatozoa swimming speed; in other words, the relationship between oxidative stress elements in the sperm and sperm quality is directly proportional. Additionally, in the same study, it was also observed that variations in spermatozoa morphology increased due to the increase in the stress environment in dominant males and hierarchically weak males. The fact that the animal material used in our study is songbirds coincides with the current study. However, in the present study, only the amount of antioxidant in the germinated wheat was determined, and its effects on some sperm parameters related to the diet provided to canaries were evaluated. The results achieved emphasized the negative effects of oxidative stress on sperm morphology. However, in the present study, there was no significant difference between the control and treatment groups. This might be because the canaries were not exposed to stress such as competition, and the study was carried out in individual cages in an equal environment for each bird. In addition, it was stated in the study that the swimming speed of spermatozoa was negatively affected by oxidative stress and slowed down. Although sperm swimming speed was not evaluated in our study, the increase in motility and viability in the treatment group supports the findings of the current study.

Wheat sprout extract is also known to be rich in zinc [28]. It was emphasized that selenium and zinc addition increased sperm motility and concentration in farm animals and poultry [78]. It was stated in a previous study that the addition of 60 mg/kg Zn-Met in diet positively affected sperm quality in roosters [79]. In another study, it was reported that zinc’s addition to diet increased the plasma testosterone level in brood pigeons [80].

## 5. Conclusions

In this study, the antioxidant activities of germinated wheat seeds were determined, and their effects on some spermatological parameters in Gloster canaries were examined.

It was determined that the antioxidant activity of wheat increased with increases in germination time. The best results, in terms of antioxidant parameters in germinated wheat, were obtained from the extracts prepared with 90% ethanol on the fifth day.

Motility and viability rate are among the most important examination parameters in the microscopic examination of spermatozoa. Motility refers to the ratio of spermatozoa moving strongly in some direction to those moving in another direction or not moving at all. High motility and vitality rate are the basic characteristics expected from quality sperm. It was observed that germinated wheat fed to birds to increase reproductive performance in canaries had a positive effect on spermatozoa motility and viability. It is thought that the main reason for this outcome is the presence of phenolic compounds in germinated wheat. Since improving sperm quality in songbirds directly increases reproductive success, it is predicted that feeds containing natural antioxidants might increase the fertilization rate in birds. Studies on sperm freezing, cryopreservation, and artificial insemination in songbirds remain up to date. The main requirement for these studies to achieve the desired results is to increase the quality of sperm. 

As a result, it can be stated that germinated seed has positive effects on some spermatological parameters in Gloster canaries. However, more comprehensive studies on this subject are needed. 

## Figures and Tables

**Figure 1 vetsci-11-00004-f001:**
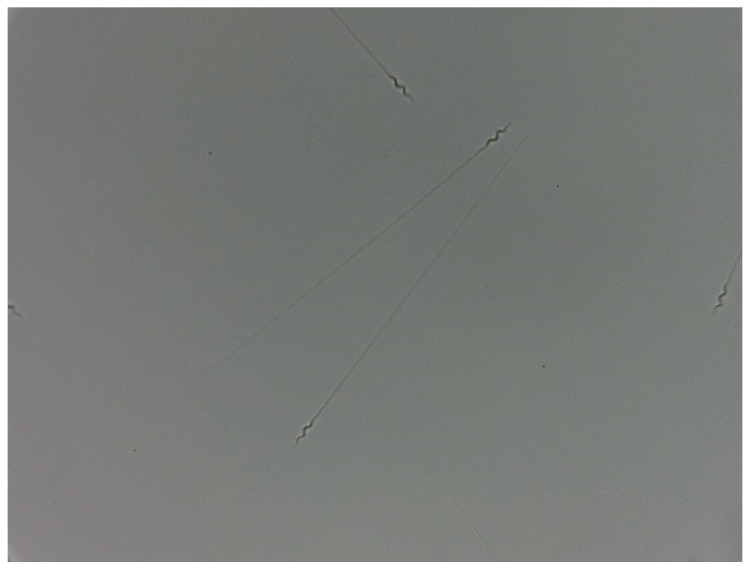
Canary spermatozoa (40× magnification).

**Figure 2 vetsci-11-00004-f002:**
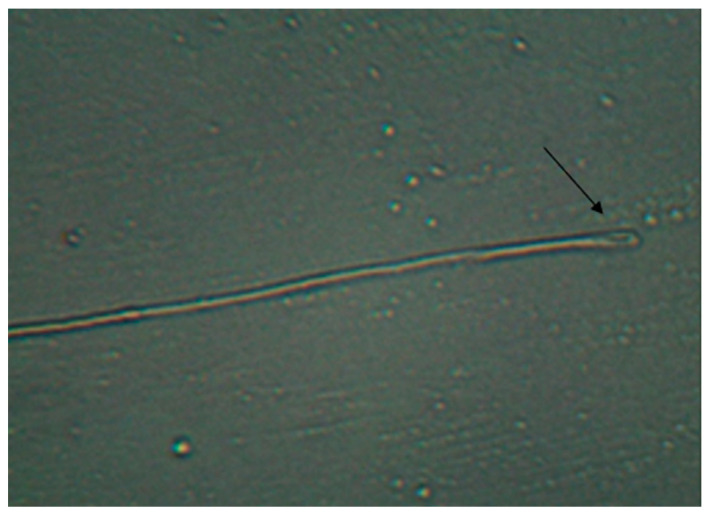
Spermatozoon tail (Hypo-osmotic Swelling Test). (Arrow indicates curling in the tail).

**Figure 3 vetsci-11-00004-f003:**
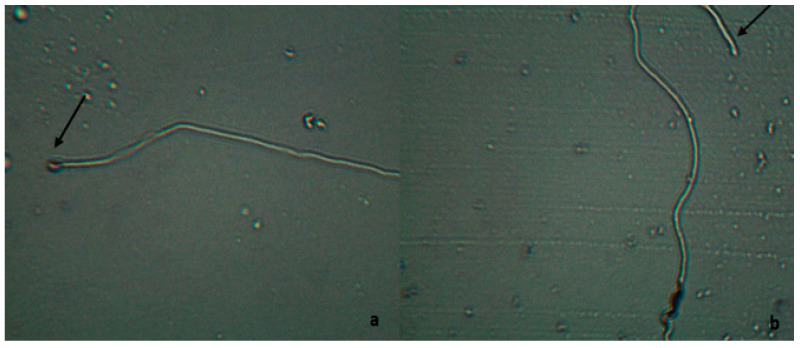
Abnormal spermatozoon (**a**,**b**). Arrow in (**a**) indicates immature acrosome. Arrow in (**b**) indicates curled tail.

**Figure 4 vetsci-11-00004-f004:**
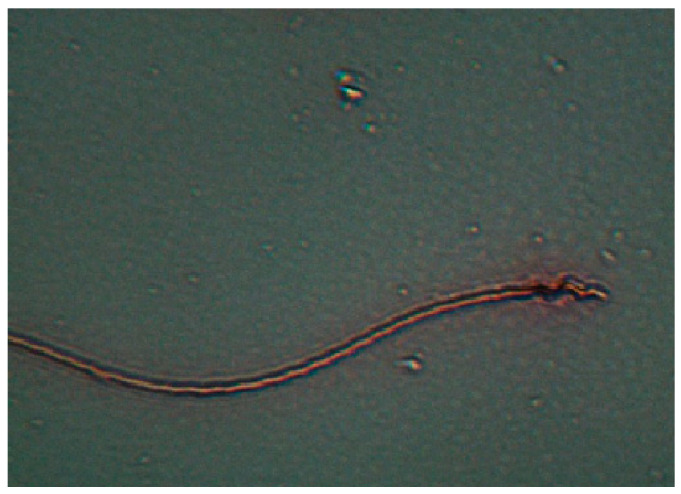
Dead spermatozoon (The eosin staining).

**Table 1 vetsci-11-00004-t001:** Experiment plan.

Groups	Number of Replicates	Germinated Wheat
C	6	−
GW	6	+

C: control; GW: group fed germinated wheat added to basal diet.

**Table 2 vetsci-11-00004-t002:** Nutritional content of germinated wheat.

Proximate Composition	Day 1	Day 3	Day 5
Dry matter (%)	91.91	92.99	92.50
Crude protein (%)	14.03	14.40	14.82
Ether extract (%)	1.60	1.82	1.80
Crude ash (%)	1.58	1.70	1.89
Crude fibre (%)	3.17	3.21	3.23

**Table 3 vetsci-11-00004-t003:** Antioxidant activity of germinated wheats.

Germination Time (day)		DPPH (%)	Total Phenolic Content mg GAE/g
1		63.09 ^c^	157.00 ^c^
3		71.63 ^a^	236.58 ^b^
5		69.23 ^b^	218.42 ^a^
*p* values		<0.001	<0.001
F values		698.65	12,737.07
Solvent rate (%)			
30		67.04 ^b^	196.17 ^c^
60		58.12 ^c^	243.42 ^a^
90		78.79 ^a^	235.42 ^b^
*p* values		<0.001	<0.001
F values		3865.75	2051.53
Germination time × Solvent rate (day/%)			
1	30	64.42 ± 0.37 ^e^	149.75 ± 2.22 ^g^
	60	56.89 ± 0.61 ^f^	136.25 ± 1.71 ^h^
	90	67.95 ± 0.52 ^d^	185.00 ± 1.16 ^f^
3	30	69.39 ± 0.81 ^c^	203.75 ± 1.71 ^e^
	60	63.14 ± 0.37 ^e^	301.00 ± 1.41 ^b^
	90	82.37 ± 0.37 ^b^	205.00 ± 1.41 ^e^
5	30	67.31 ± 0.74 ^d^	235.00 ± 2.45 ^d^
	60	54.33 ± 0.61 ^g^	293.00 ± 2.94 ^c^
	90	86.06 ± 0.61 ^a^	316.25 ± 1.71 ^a^
*p* values		<0.001	<0.001
F values		359.93	1913.28

a–h: The averages with different superscripts in the same column differ significantly (*p* < 0.05).

**Table 4 vetsci-11-00004-t004:** Spermatological parameters in canaries.

Groups	Motility (%)	Abnormal Sperm (%)	HOST (%)	Viability Rate (%)
C	42.50 ± 3.59	25.50 ± 2.41	75.00 ± 2.52	46.83 ± 4.53
GW	55.83 ± 3.96	26.67 ± 2.84	79.67 ± 2.42	59.17 ± 2.80
*p* values	0.032	0.760	0.211	0.043

C: control (basal diet); GW: group fed germinated wheat added to the basal diet; HOST: hypo-osmotic swelling test.

## Data Availability

The data analyzed for the study are available from the corresponding author upon reasonable request.

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
