# Peer review of "Germinated Wheat as a Potential Natural Source of Antioxidants to Improve Sperm Quality: A Canary Trial"

_vetsci, 2023, doi:10.3390/vetsci11010004_

Round 1

Reviewer 1 Report

Comments and Suggestions for Authors

It is a well-written manuscript that addresses a topical issue: sperm quality or male fertility and antioxidants. Even if I doubt that artificial insemination is important for the breeding of canaries, I think the question of the study is relevant.

The manuscript has two mistakes that should be corrected before publication:

- in the germinated wheat are not only antioxidants but also selenium and zinc. These ingredients can also have an influence on sperm quality. The authors address this aspect briefly in the discussion without going into depth.

- There is a lack of current literature (e.g. Golzar Adabi S, Karimi Torshizi MA, Raei H, Marnewick JL. Effect of dietary n-3 fatty acid and rooibos (Aspalathus linearis) supplementation on semen quality, sperm fatty acids and reproductive performance of aged male broiler breeders.  J Anim Physiol Anim Nutr (Berl). 2023 Jan;107(1):248-261).

A lot of research has been done on this topic during the last three years. Please add to and discuss the current literature.

Author Response

REVISION

Dear Editor,

All of the corrections were painted red in the text. And the corrections were given under the title of “RESPONSE to REVIEWERS”.

First of all, I would like to express my gratitude to the Reviewers for their valuable contributions. I have found the Reviewers extremely sensitive about reading and examining the article. I present the corrections as I have made taking the criticism of the Reviewers into consideration as follows.

RESPOND TO REVIEWERS

REVIEWER 1

All of the corrections were painted red in the text. And the corrections were given under the title of “RESPONSE to REVIEWERS”.

The manuscript has two mistakes that should be corrected before publication:

  1. ‘- in the germinated wheat are not only antioxidants but also selenium and zinc. These ingredients can also have an influence on sperm quality. The authors address this aspect briefly in the discussion without going into depth.’

Revision/Answer: Thank you for your evaluation. The additions were made to the discussion in terms of selenium and zinc.

  1. There is a lack of current literature (e.g. Golzar Adabi S, Karimi Torshizi MA, Raei H, Marnewick JL. Effect of dietary n-3 fatty acid and rooibos (Aspalathus linearis) supplementation on semen quality, sperm fatty acids and reproductive performance of aged male broiler breeders. J Anim Physiol Anim Nutr (Berl). 2023 Jan;107(1):248-261). A lot of research has been done on this topic during the last three years. Please add to and discuss the current literature.

Revision/Answer: Thank you for your evaluation. The current sources were added for discussion.

OTHERS:

The manuscript was checked again in terms of Language.

Reviewer 2 Report

Comments and Suggestions for Authors

The experimental design is correct but the following comments need to be addressed:

Experimental plan and feeding of the canaries.

It would be informative that the authors provide data about the mean weight of each canary group before and after the trial to verify this parameter does not affect the outcome. Also, details like how the germinated wheat was fed to the canaries are important as well as the amount of supplement provided and the way its ingestion was confirmed. The length of the treatment and other husbandry parameters mentioned in the abstract (photoperiod, water) should be also indicated in this section. 

Collecting sperm from canaries 

Were sperm diluted after collection? If yes, which extender was used?

Evaluation of spermatological parameters

When the authors mention that the motility was recorded, did they use any computerized system to record precise values? Figure 1 does not show the percentage of motility. For Figures 2 and 3 please indicate the abnormalities with arrows. Also, please insert the magnification value in all figures. Which were the mean sperm concentration and volume at each group? The sperm quality parameters before the dietary treatment need also be stated.

Statistical analysis

Were data tested for normality and homogeneity of variance?

Results

Table 3. Substitute slashes (/) in column 1 for hyphens (-). How many wheat samples were used at each analysis of the antioxidant activity? Whys isn't the SEM (standard error of the mean) indicated for the total phenolic content as in the DPPH? Also, the SEM must be indicated in Table 4 (sperm quality parameters).

Discussion 

Line 283. Selenium content was not measured so it is not a valid conclusion that selenium affects spermatozoa.

Conclusions

Line 328. Motility does not exclusively relate to forward movement because sperm with circular trajectories are still considered to be motile. 

Line 339. "wheat germ has a supportive effect on fertilisation success in Gloster canaries". However, fertility data is not provided. 

Comments on the Quality of English Language

The manuscript reads well except in some passages. Some typographical mistakes have been also detected.

Author Response

Manuscript Title: Germinated Wheat as a Potential Natural Source of Antioxidants to Improve Sperm Quality: A Canary Trial

REVISION

Dear Editor,

All of the corrections were painted red in the text. And the corrections were given under the title of “RESPONSE to REVIEWERS”.

First of all, I would like to express my gratitude to the Reviewers for their valuable contributions. I have found the Reviewers extremely sensitive about reading and examining the article. I present the corrections as I have made taking the criticism of the Reviewers into consideration as follows.

RESPOND TO REVIEWERS

REVIEWER 2

All of the corrections were painted red in the text. And the corrections were given under the title of “RESPONSE to REVIEWERS”.

The experimental design is correct but the following comments need to be addressed:

  1. ‘Experimental plan and feeding of the canaries.’

(1/1: It would be informative that the authors provide data about the mean weight of each canary group before and after the trial to verify this parameter does not affect the outcome.)

Revision/Answer (1/1): Thank you for your evaluation. The canary weights were not determined before and after the experiment in this study. Determination of the weight of canaries is a very difficult process and there is no correlation with weight in terms of the parameters examined. Canaries with similar characteristics were selected while forming the groups at the beginning of the experiment. The homogeneity of variances in the groups were tested in terms of parameters such as motility.

(1/2: Also, details like how the germinated wheat was fed to the canaries are important as well as the amount of supplement provided and the way its ingestion was confirmed. The length of the treatment and other husbandry parameters mentioned in the abstract (photoperiod, water) should be also indicated in this section.)

Revision/Answer (1/2): Thank you for your evaluation. The requested additions were made to the section on the experimental plan and feeding of the canaries.

  1. ‘Collecting sperm from canaries’

(Were sperm diluted after collection? If yes, which extender was used?)

Revision/Answer (2):  Thank you for your evaluation. Requested addition was made. (The sperm samples taken from the canaries at the end of the experiment were examined without any delay. DMEM diluter was used while analyzing the fresh sperm motility parameter.)

3- ‘Evaluation of spermatological parameters’

(3/1: When the authors mention that the motility was recorded, did they use any computerized system to record precise values?)

Revision/Answer (3/1): Thank you for your evaluation. No. No computerized system was used for motility. But motility assessments were done by the same person.

(3/2: Figure 1 does not show the percentage of motility.)

Revision/Answer (3/2): Thank you for your evaluation. Figure 1 illustrates the spermatozoon at x40 magnification during the motility analysis. An image containing intense spermatozoa was avoided to achieve a clear image.   

(3/3: For Figures 2 and 3 please indicate the abnormalities with arrows. Also, please insert the magnification value in all figures.)

 Revision/Answer (3/3): Thank you for your evaluation. Addition was made on Figure 2 and Figure 3.

(3/4: Which were the mean sperm concentration and volume at each group? The sperm quality parameters before the dietary treatment need also be stated.)

 Revision/Answer (3/4): Thank you for your evaluation. The homogeneity of variances in the groups were tested in terms of parameters such as motility (P>0.05).

4- ‘Statistical analysis’

(4: Were data tested for normality and homogeneity of variance?)

Revision/Answer (4): Thank you for your evaluation. Yes, a normality test was conducted, and it was determined that the data had a normal distribution.

5- ‘Results’

[5/1: Table 3. Substitute slashes (/) in column 1 for hyphens (-)]

Revision/Answer (5/1): Thank you for your evaluation. It was corrected as slashes (/)

(5/2: How many wheat samples were used at each analysis of the antioxidant activity?)

Revision/Answer (5/2): Addition was made. (Antioxidant activity in germinated wheat was examined with four replications.)

(5/3: Whys isn't the SEM (standard error of the mean) indicated for the total phenolic content as in the DPPH?)

Revision/Answer (5/3): Thank you for your evaluation. Addition was made.

[5/4: Also, the SEM must be indicated in Table 4 (sperm quality parameters)]

Revision/Answer (5/4): Thank you for your evaluation. Addition was made.  

6- ‘Discussion’

(6: Line 283. Selenium content was not measured so it is not a valid conclusion that selenium affects spermatozoa.)

Revision/Answer (6): Germinated wheat is known to be rich in selenium. It is given in introduction section of manuscript. Comments were made based on this information.

7- ‘Conclusions’

(7/1: Line 328. Motility does not exclusively relate to forward movement because sperm with circular trajectories are still considered to be motile.)

Revision/Answer (7/1): Necessary correction was made in manuscript.

(7/2: Line 339. "wheat germ has a supportive effect on fertilisation success in Gloster canaries". However, fertility data is not provided.)

Revision/Answer (7/2): Necessary correction was made in manuscript.

OTHERS:

The manuscript was checked again in terms of Language.

Reviewer 3 Report

Comments and Suggestions for Authors

Dear Authors,

I attach a review of the article „ Germinated Wheat as a Potential Natural Source of Antioxidants to Improve Sperm Quality: A Canary Trial”.

MAIN NOTES

·        Not all results have been confirmed by statistical analyses. Necessary statistical analyzes in the result part

·        Section Material and Methods must be improved; comments below

Other comments:

Line 67: …Wheats were germinated…

Rev: a short characterization should be given; source of origin, species, variety, etc.

Line 73-74: …this way, the germination process is completed…

Rev: how long it took to germinate?; how old the sprouts were? Information should be given in manuscript.

Line 102: …formed, with 6 male Gloster canaries the age of 2-3 years in each group (Table 1)…

Rev: latin name should be given.

Line 162: Table 3. Antioxidant activity of germinated wheats

Rev: Total phenolic content should be given as mean±SD. Adequate statistic tool should be use to show statistically differences, where Germination time and Solvent rate are two factors.

Line 164-166: …The highest DPPH reduction % value (86.06%) and total phenolic content were observed on the 5th day of germination and in the extracts prepared with 90% ethanol. This result reveals that antioxidant capacity increases in parallel with the increase in germination time…

Rev: Adequate statistic tool should be use to show statistically differences, where Germination time and Solvent rate are two factors.

Line 177, 220: …(P<0.05)…

Rev: should be (p<0.05).

References

Line 413-414: …Morovvati, H.; Adibmoradi, M.; Sheybani, M. T.; Salar Amoli, J.S.A. Effects of Wheat Sprout Extract on the Quality of Sperm in Rats Exposed to Lead. Iran. Vet. J. 2016, 12, 76-85…

Rev: formatting – unification.

Line 422-423: …var. polypodiifolia…

Rev: polypodiifolia should be in italics.

Line 422-423: …Fron.t Zoo…

Rev: ?

Line 433: …Tachycineta bicolor…

Rev: Tachycineta bicolor should be in italics.

Line 440: …Falco…

Rev: Falco should be in italics.

Line 484: Žilić, S.; Basi´c, Z.; Hadži-Taškovi´c Šukalovi´c, V.; Maksimovi´c, V.; Jankovi´c, M.; Filipovi´c, M …

Rev: should be corrected.

Line 519: …H.A…

Rev: space.

Line 522: …Passer domesticus…

Rev: Passer domesticus should be in italics

Rev: In vitro in all manuscript should be in italics.

Comments on the Quality of English Language

n

Author Response

Manuscript Title: Germinated Wheat as a Potential Natural Source of Antioxidants to Improve Sperm Quality: A Canary Trial

REVISION

Dear Editor,

All of the corrections were painted red in the text. And the corrections were given under the title of “RESPONSE to REVIEWERS”.

First of all, I would like to express my gratitude to the Reviewers for their valuable contributions. I have found the Reviewers extremely sensitive about reading and examining the article. I present the corrections as I have made taking the criticism of the Reviewers into consideration as follows.

RESPOND TO REVIEWERS

REVIEWER 3

All of the corrections were painted red in the text. And the corrections were given under the title of “RESPONSE to REVIEWERS”.

MAIN NOTES

  1. ‘Not all results have been confirmed by statistical analyses. Necessary statistical analyzes in the result part.’

Revision/Answer (1/1): Thank you for your evaluation. Additions were made.

  1. ‘OTHER COMMENTS’

(Section Material and Methods must be improved; comments below)

[2/1: Line 67: …Wheats were germinated…]   (Rev: a short characterization should be given; source of origin, species, variety, etc.)

Revision/Answer (2/1): Thank you for your evaluation. The information was given about source of origin, species, variety, etc.

[2/2: Line 73-74: …this way, the germination process is completed… (Rev: how long it took to germinate?; how old the sprouts were? Information should be given in manuscript.)

Revision/Answer (2/2): Thank you for your evaluation. The information was given in manuscript about wheat germination.

[2/3: Line 102: …formed, with 6 male Gloster canaries the age of 2-3 years in each group (Table 1)…] (Rev: latin name should be given.)

Revision/Answer (2/3): Thank you for your evaluation. Latin name of Canary name was added as Serinus canaria.

[2/4: Line 162: Table 3. Antioxidant activity of germinated wheats] (Rev: Total phenolic content should be given as mean±SD. Adequate statistic tool should be use to show statistically differences, where Germination time and Solvent rate are two factors.)

Revision/Answer (2/4): Thank you for your evaluation. Germination time and solvent rate were made as two factors statistic. Standart errors were given in Table.

[2/5: Line 164-166: …The highest DPPH reduction % value (86.06%) and total phenolic content were observed on the 5th day of germination and in the extracts prepared with 90% ethanol. This result reveals that antioxidant capacity increases in parallel with the increase in germination time…] (Rev: Adequate statistic tool should be use to show statistically differences, where Germination time and Solvent rate are two factors.)

Revision/Answer (2/4): Thank you for your evaluation. It was made.

[2/6: Line 177, 220: …(P<0.05)…] [Rev: should be (p<0.05).]

Revision/Answer (2/3): Thank you for your evaluation. However, the p value, which is the level of significance in statistics, can also be written in uppercase letters.

OTHERS:

The manuscript was checked again in terms of Language.

Round 2

Reviewer 3 Report

Comments and Suggestions for Authors

Dear Authors,

I attach a review (second step) of the article „ Germinated Wheat as a Potential Natural Source of Antioxidants to Improve Sperm Quality: A Canary Trial”.

General remarks

2.7. Statistical Analysis

Rev: Inappropriate statistical analysis!

Rev: The method of analysis is not acceptable. Antioxidant activity of germinated wheats should be analysed using two-way ANOVA, and post-hoc test (e.g. Tukey test) must be used. Germination time is the first factor (3 levels – 1, 3, 5, day) and solvent rate is the second factor (3 levels – 30, 60, 90%). F-values and p-values should be given in manuscript for 1 factor (germination time), 2 factor (solvent rate) and their interaction (germination time x solvent rate) as a results of two-way ANOVA.

Table 3

Rev: The level of significance should be presented as “p”

Rev: In Table 3 is P=0.000, that's what the program showed. Should be: p<0.001.

Table 4. … 42.50±3.594 …

Rev: 3.594 –  Entering SD with an accuracy of thousandths is not necessary - it does not contribute anything.

Rev: Is P values; should be p-values.

Author Response

RESPOND TO REVIEWERS

1.

2.7. Statistical Analysis

Rev: Inappropriate statistical analysis!

Rev: The method of analysis is not acceptable. Antioxidant activity of germinated wheats should be analysed using two-way ANOVA, and post-hoc test (e.g. Tukey test) must be used. Germination time is the first factor (3 levels – 1, 3, 5, day) and solvent rate is the second factor (3 levels – 30, 60, 90%). F-values and p-values should be given in manuscript for 1 factor (germination time), 2 factor (solvent rate) and their interaction (germination time x solvent rate) as a results of two-way ANOVA.

Revision/Answer (1): Thank you for your evaluation. The corrections were made. Additions were made for Abstract and Statistic Analysis in Method. When the interaction was made in the general linear model, the standard deviation for time and concentration could not be added because it did not appear in SPPS. Therefore, only standard deviations of the germinated time x concentration interaction were given. Moreover, F values were added in Table.

2.

Table 3

Rev: The level of significance should be presented as “p”

Rev: In Table 3 is P=0.000, that's what the program showed. Should be: p<0.001.

Revision/Answer (2): Thank you for your evaluation. It was corrected.

3.

Table 4. … 42.50±3.594 …

Rev: 3.594 –  Entering SD with an accuracy of thousandths is not necessary - it does not contribute anything.

Revision/Answer (3): Thank you for your evaluation. It was corrected.

 Rev: Is P values; should be p-values.

Revision/Answer (4): Thank you for your evaluation. It was corrected.

Round 3

Reviewer 3 Report

Comments and Suggestions for Authors

Line 177-178: …General Linear Model procedure (Two-Way ANOVA)…

Rev: should be: Two-Way ANOVA …

Table 3

Rev:  – p value = 0.000 it is impossible; it is program information. Should be p<0.001.

Rev: 86.06±0.61a – font size

Author Response

Manuscript Title: Germinated Wheat as a Potential Natural Source of Antioxidants to Improve Sperm Quality: A Canary Trial

REVISION

Dear Editor,

All of the corrections were painted red in the text. And the corrections were given under the title of “RESPONSE to REVIEWER”.

First of all, I would like to express my gratitude to the Reviewers for their valuable contributions. I have found the Reviewers extremely sensitive about reading and examining the article. I present the corrections as I have made taking the criticism of the Reviewers into consideration as follows.

RESPOND TO REVIEWERS

1.

Line 177-178: …General Linear Model procedure (Two-Way ANOVA)…

Rev: should be: Two-Way ANOVA …

 Revision/Answer (1): Thank you for your evaluation. The corrections were made.

Table 3

Rev:  – p value = 0.000 it is impossible; it is program information. Should be p<0.001.

Revision/Answer (2): Thank you for your evaluation. The corrections were made.

Rev: 86.06±0.61a – font size

Revision/Answer (3): Thank you for your evaluation. The corrections were made.
